# OpenReview forum: "BEYOND RETRIEVAL: GENERATIVE EVIDENCE CALIBRATION FOR ANSWER-UTILITY SEARCH"
_ICLR.cc/2026/Conference — ICLR 2026 Conference Withdrawn Submission_

### Official Review · Reviewer_m1aS · 2025-10-25

**Soundness:** 2
**Presentation:** 1
**Contribution:** 2
**Rating:** 2
**Confidence:** 3

**Summary:**

This paper proposes ranking retrieval results according to how much they can provide evidence for answer generation (Generative Evidence Calibration), in addition to relevance. Though the idea is great, the method does not seem novel and the method is not clearly described.

**Strengths:**

S1. Considering how much a retrieval result would provide evidence to generation seems reasonable.

S2. The paper has presented analysis and discussion regarding the key ideas.

**Weaknesses:**

W1. It is unclear how exactly GES is measured. For example, why do we count citations to a document, why does it work, and why does it work better than other methods of computing "importance"? Similarly, how is Best_doc indicator computed? Why does the weighted sum work to determine if the retrieval results provide evidence to the generation?

W2. What if high-citation documents do not directly answer the question? Even worse, what if it provides out-of-date information, or provides information of entities of confusing names?

W3. The improvement shown in Table 2 is fairly small. It is hard to understand how much is coincidence and how much is real improvement.

W4. Study is needed for different ways of counting citations. Also, why is calibration needed? Can we skip it?

**Questions:**

Please explain questions listed in the weaknesses.

---

### Official Review · Reviewer_59Wx · 2025-10-30

**Soundness:** 2
**Presentation:** 2
**Contribution:** 3
**Rating:** 4
**Confidence:** 2

**Summary:**

The paper addresses an important problem of information retrieval systems: at some point optimization of query-document similarity ceases to improve answer utility. Authors propose Generative Evidence Calibration (GEC), which measures how documents contribute to synthesized answers rather than how they match queries. GES signal is fused with BM25 and BGE retrieval scores either via learned weighted reciprocal rank fusion (GEC-WRRF) or a guarded product-of-experts (gPoE-HeadSafe) multiplier. GEC is tested across 5 datasets and multiple models (from 3.8B to 70B parameters, + Claude 3.5 Sonnet), and provides consistent gains over strong baselines such as BM25+BGE. Authors also find that cross-encoder reranking harms GEC performance by downplaying the evidence-based signal.

**Strengths:**

Exceptionally clear and concise description of the core problem.

Sound evaluation setup including oracle upper bound analysis and paired bootstrap significance testing.

Proposed methods deliver strong results across 5 diverse datasets.

Methods are ablated, and all components are shown to contribute to overall quality.

Findings are tested across models (Table 4) and domains (Table 2, 3).

**Weaknesses:**

The paper does not clearly state how key hyperparameters were set. Equation (1) uses parameters alpha and beta for citation count vs best-doc weight, but their values are not given in the main text. Ranges for gPoE hyperparameters are provided in Table 1, but exact values are not specified, and impact of these hyperparameters on performance is not clarified.

WRRF training procedure is unclear — what is the training set?

Abstract is overloaded with abbreviations and hard to interpret without domain expertise.

Table 4 does not provide standard deviation or other measure of spread, despite small sample size (100 queries). Uniform improvement suggests the results are significant, however it’s better to add some error bars / confidence intervals for maximum rigour.

It would also be interesting to see BM25+BGE (RRF) in Table 4.

Main text is often not clearly written:
Related work, 087-089 — this would benefit from elaboration. What is identification in this context? What exactly is citation generation?

How it is different from evidence calibration?

107 “Key configuration knobs: packs = 1–12,13–40,41–100,” is unclear.

123-125 “Critical implementation detail” is unclear without context.

**Questions:**

What alpha and beta were used in Multi-GEC for each dataset?
What hyperparameters were used for Guarded Product-of-Experts for each dataset?
What dataset was used to train WRRF?
These are critical questions, which significantly affect my overall score, which would be higher otherwise.

---

### Official Review · Reviewer_5Hzh · 2025-10-30

**Soundness:** 2
**Presentation:** 1
**Contribution:** 2
**Rating:** 2
**Confidence:** 5

**Summary:**

The paper proposes Generative Evidence Calibration (GEC): it derives evidence signals from citation-anchored “synthesis packs” (Multi-GES) and fuses them with BM25/BGE using either a guarded product-of-experts (gPoE-HeadSafe) or a learned WRRF fusion.

**Strengths:**

1. Reframing retrieval around answer utility. The paper clearly positions evidence utility—not similarity—as the ranking target and operationalizes it via a three-stage pipeline (portfolio synthesis → Multi-GES → calibrated fusion).

2. Diagnostic headroom analyses and cost accounting. The Oracle Upper Bound / PoE Reachability framing is a useful diagnostic lens, and the production appendix includes back-of-the-envelope cost comparisons.

**Weaknesses:**

1. Writing quality and formulation clarity are below venue bar. The prose is frequently rhetorical (“fundamental paradigm crisis… inevitable future”), which hurts readability and obscures the algorithmic core. More importantly, key definitions are either informal or split across figures and claims: Figure 1 only sketches “Input: Query + Retrieved Documents” without a precise problem statement; WRRF “gate features” are named but not fully specified; and APC is invoked as a central component in the abstract yet receives no algorithmic definition or equation in the main body, leaving a gap between claims and method.

2. No serious treatment of query decomposition or multi-round search; advantage over iterative methods is unsubstantiated. Although iterative retrieval is acknowledged in related work, the method fixes a single candidate pool (BM25/BGE) and performs portfolio generation on that pool—there is no query planning/decomposition, no iterative expansion/verification, and no comparison against modern agentic/self-reflective retrieval frameworks (e.g., Self-RAG-style pipelines). Consequently, it’s unclear whether GEC’s gains persist once competitive iterative search is in play.

3. Baselines are dated/incomplete and the evaluation protocol weakens comparability. Main tables compare to BM25, BGE, and BM25+BGE (RRF) only; state-of-the-art rerankers/multi-vector retrievers (e.g., monoT5-class, SPLADE/ColBERT-family, or strong E5/BGE-rerankers) are absent from the headline results (a “BGE-Reranker” appears only in a cost aside). Further, results are reported on small random slices (≥50 topics, 3 seeds), which reduces comparability to standard benchmarks and risks instability. Overall, the evidence does not establish superiority over the latest search methods.

**Questions:**

refer to weakness

---

### Official Review · Reviewer_2QmN · 2025-10-31

**Soundness:** 2
**Presentation:** 1
**Contribution:** 2
**Rating:** 2
**Confidence:** 4

**Summary:**

This paper introduces Generative Evidence Calibration (GEC), a framework for improving document retrieval in knowledge-intensive tasks like RAG by aligning ranking with evidence utility rather than raw similarity. The approach builds a “portfolio” of LLM-generated answers with sentence-level citations (from the pool of retrieved documents), extracts a Multi-GES (==multi-granular evidence score) from those citations, and combines this signal with BM25 and dense retrieval (BGE) scores through two fusion mechanisms: a Weighted Reciprocal Rank Fusion (WRRF) model and a Guarded Product-of-Experts (gPoE-HeadSafe) mechanism.

The authors evaluate on multiple datasets (Natural Questions, FiQA, SciFact, HotpotQA, TREC-COVID) and show **moderate** gains in MRR@10 and recall over BM25+BGE baselines, while claiming that cross-encoder rerankers actually hurt calibrated retrieval.

**Strengths:**

* The studied problem is conceptually interesting: standard ranking systems (retrieval or reranking) optimize for relevance (usually derived from human judgements), but not actual utility, i.e., the extent to which the document is useful to answer the question (== “how documents contribute to synthesized answers rather than how they match queries”). This is particularly interesting for RAG systems which have become standard in industry. However, previous (missing) works have studied similar topics (see weaknesses).
* Empirical rigor: overall, authors evaluate their approach across five datasets and multiple LLMs, with detailed tables and reproducibility notes in Appendix. It is interesting to see that the method which does not rely on relevance judgements (but rather citation extraction from LLMs) actually works on standard IR benchmarks (with relevance annotation).
* Useful negative finding: the observation that cross-encoder reranking can degrade calibrated evidence quality (Section 6.2) is counterintuitive and valuable, and further highlights the motivation of the paper.

**Weaknesses:**

My main overall concern lies in the scope and clarity of the paper. To me, it reads more like a technical report focused on industrial system improvements rather than a genuine machine learning contribution suitable for ICLR. The work might be a better fit for conferences such as SIGIR or EMNLP. Below, I provide a more detailed discussion of specific concerns.

* Presentation/clarity: Overuse of acronyms (GES, PRA, WRRF, OUB, etc.) makes sections dense and hard to follow => the abstract for instance is borderline understandable. Some methodological details are buried in appendices or informal prose rather than formal equations. Also, some parts feel more like a blogpost; very short and not detailed (for instance “Critical implementation detail:” at the end of Section 2.3, or the “Efficiency” part in Section 4). The section “Guarded Product-of-Experts” (Section 3.3) contains a lot of details without any context (max_jump, lambda_cap etc.). etc. In addition, the use of certain terms (such as “Portfolio synthesis” or “Product-of-Experts”) feels somewhat overloaded and, in my opinion, adds unnecessary confusion. Why is the ColBERT paper referenced at the end of the sentence “new retrieval paradigm using Product-of-Experts calibration” (Section 2)?
* Overall, a lot of missing details: it was initially hard for me to understand how the Portfolio synthesis works: i) what are packs? ii) what exactly are query rewrites? iii) what is m? I was only able to make educated guesses. For instance, Figure 5/Figure 6 suggest different “Portfolio variant accumulations” without any description (and so m=5, according to the definition of w_i); but what are “causal” or “risk” variants? Details about the learning-to-rank approach (LightGBM) are also very limited; e.g., what’s the training signal? What is “Traditional” in Eq. 2? etc…
* Given the motivation for the paper (optimizing for utility), I have the feeling that it would have been more helpful to evaluate the approach in a RAG setting rather than a pure IR task (all metrics are retrieval-level, i.e., MRR@10 or nDCG). It does not necessarily invalidate the approach, but it would strengthen the results.
* Gains **seem numerically small** and not clearly statistically significant. The gPoE variant is sometimes worse than WRRF and occasionally below BM25+BGE (RRF).
* I think it is necessary to add results from both a standard BGE + cross-encoder and BGE + LLM reranker into Table 2 and Table 3. Because GEC-WRRF combines BM25, BGE dense retrieval, and Multi-GES, you can only expect gains compared to BM25+BGE no? As authors state (Section 4), “Portfolio generation replaces quadratic pairwise reranking”, so it feels natural to include those results in main Tables. To my understanding, cross-encoders are studied in Section 6 only on top of the existing method (Portfolio PoE vs Portfolio PoE+CE), but not as a direct competitive comparison.
* I was a bit confused by the OUB/PRA metrics which feel opaque and self-referential => bounded by the limited document pool, not an absolute notion of oracle performance etc.
* Some related works are missing: many works have explored how to use LLM feedback to better align retrievers (e.g., [1,2]) for generation, and recent works have directly explored the notion of utility of documents [3,4] in RAG. None of them are discussed in this paper.

[1] REPLUG: Retrieval-Augmented Black-Box Language Models
[2] Augmentation-Adapted Retriever Improves Generalization of Language Models as Generic Plug-In
[3] Training a Utility-based Retriever Through Shared Context Attribution for Retrieval-Augmented Language Models
[4] Leveraging LLMs for Utility-Focused Annotation: Reducing Manual Effort for Retrieval and RAG

**Questions:**

See questions in ’Weaknesses’ section; also:

* How accurate are LLM citations? Did you manually verify whether sentences correctly attribute to cited passages? What is the error rate across datasets? You also examine whether citation frequency correlates with document length, lexical overlap etc.

* Can you clarify which cross-encoders were tested (architecture, training corpus, cutoff depth)? Have you tried re-training a cross-encoder to predict evidence utility (GES) rather than relevance to verify if the degradation persists? Could you achieve similar results by learning a supervised utility predictor instead of relying on LLM-generated citations?

---

### Note · Authors · 2025-11-12

**Comment:**

We appreciate the reviewers’ time and feedback. After consideration, we have chosen to withdraw this submission to refine the framing and extend the evaluation toward a venue better aligned with its focus on retrieval-utility modeling and evidence calibration. The next version will use the current framework to do a wider analysis of retrieval and synthesis paradigms.

**Withdrawal Confirmation:**

I have read and agree with the venue's withdrawal policy on behalf of myself and my co-authors.